# Decreasing Trends of Secondary Primary Colorectal Cancer among Women with Uterine Cancer: A Population-Based Analysis

**DOI:** 10.3390/jcm8050714

**Published:** 2019-05-20

**Authors:** Koji Matsuo, Rachel S. Mandelbaum, Hiroko Machida, Kosuke Yoshihara, Franco M. Muggia, Lynda D. Roman, Jason D. Wright

**Affiliations:** 1Division of Gynecologic Oncology, Department of Obstetrics and Gynecology, University of Southern California, Los Angeles, CA 90033, USA; rachel.mandelbaum@med.usc.edu (R.S.M.); lroman@med.usc.edu (L.D.R.); 2Norris Comprehensive Cancer Center, University of Southern California, Los Angeles, CA 90033, USA; 3Department of Obstetrics and Gynecology, Tokai University School of Medicine, Kanagawa 259-1193, Japan; hiroko.machida@tokai.ac.jp; 4Department of Obstetrics and Gynecology, Niigata University School of Medicine, Niigata 951-8510, Japan; yoshikou@med.niigata-u.ac.jp; 5Division of Medical Oncology, Department of Medicine, New York University, New York, NY 10016, USA; franco.muggia@nyulangone.org; 6Division of Gynecologic Oncology, Department of Obstetrics and Gynecology, Columbia University College of Physicians and Surgeons, New York, NY 10032, USA; jw2459@cumc.columbia.edu

**Keywords:** uterine cancer, colorectal cancer, secondary cancer, trend, survival

## Abstract

The current study examined trends, characteristics, and outcomes of women with uterine cancer who had secondary colorectal cancer. This is a retrospective study utilizing the Surveillance, Epidemiology, and End Results Program between 1973–2013. Among uterine cancer (*n* = 246,272) and colorectal cancer (*n* = 421,312) cohorts, women with both diagnoses were identified, and clinico-pathological factors and survival were extracted and analyzed. There were 6862 women with both cancer diagnoses, representing 2.8% of the uterine cancer cohort and 1.6% of the colorectal cancer cohort. Among 123,940 women with uterine cancer survivors, the number with postcedent colorectal cancer decreased from 5.3% to 0.7% between 1981–2008 (relative risk reduction 87.0% *p* < 0.001). Similarly, of 141,801 women with colorectal cancer survivors, the number with postcedent uterine cancer decreased from 1.7% to 0.5% between 1973–2008 (relative risk reduction 71.6%, *p* < 0.001). In the uterine cancer cohort, women with antecedent/synchronous colorectal cancer had more high-grade tumors and advanced-stage disease resulting in poorer survival, whereas those who had postcedent colorectal cancer had more low-grade tumors and early-stage disease resulting in superior survival compared to those without secondary colorectal cancer (all, *p* < 0.05). In conclusion, the development of postcedent colorectal cancer following uterine cancer has decreased in recent years in the United States.

## 1. Introduction

Colorectal cancer and uterine cancer are the third and fourth most common female malignancies in the United States, respectively [1]. In 2019, nearly 62,000 women are estimated to be diagnosed with uterine cancer and more than 67,000 women are estimated to be diagnosed with colorectal cancer [1]. The incidence of uterine cancer has been increasing over past decades, with obesity and aging as key risk factors for uterine cancer development [2,3]. It has also been established that women with uterine cancer are at increased risk of developing secondary colorectal cancer [4,5,6,7,8,9].

This increased risk of colorectal cancer in women with uterine cancer is likely due to similar epidemiological and genetic risk factors for the two malignancies and the fact that treatments for uterine cancer such as pelvic irradiation may also increase the risk of colorectal cancer [10,11,12]. In addition, approximately 3–5% of all women with uterine cancer have Lynch syndrome, a familial cancer syndrome involving a DNA mismatch repair gene deficiency, which increases the risk of both uterine cancer and colorectal cancer [13,14].

Given recent demographic trends in the U.S. population that may impact the incidence of both uterine and colorectal cancers, namely increasing age and obesity rates as well as increasing utilization of colorectal cancer screening and surveillance in at-risk populations [15,16,17,18], it would be of value to examine population-based statistics of secondary colorectal cancer in women with uterine cancer. Moreover, the survival of women with uterine cancer who develop secondary colorectal cancer has not been examined completely [8]. The objective of the study was to examine the trends, characteristics, and outcomes of women with uterine cancer who had secondary colorectal cancer by utilizing a population-based tumor registry in the United States.

## 2. Materials and Methods

### 2.1. Study Population and Eligibility

This retrospective study examined the National Cancer Institute’s Surveillance, Epidemiology, and End Results program. This database, launched in 1973, is the largest population-based tumor registry in the United States, covering approximately 34.6% of the population in the latest version [19]. Patient identification, data entry, and rigorous quality control for this database are performed by registered and trained personnel by the National Cancer Registrars Association and the program [20]. The Institutional Review Board at the University of Southern California exempted this study due to the use of publicly available de-identified data.

The cases between 1973–2013 were obtained from the SEER*Stat 8.3.2 (IMS Inc., Calverton, MD, USA). First, two datasets were generated, one for uterine cancer and one for colorectal cancer, limited to malignancy and female sex. Cases with the same study identification number assigned by the program in each dataset were then identified and linked, and the multiple entry cases were excluded to retain only the primary entry. These cases identified in both uterine and colorectal cancer databases were considered to be secondary primary cancer (SPC) cases as described previously [21,22].

### 2.2. Clinical Information

From the uterine cancer database, the following variables were abstracted: (i) patient demographics (age, year and month at diagnosis, race, marital status, and registration area), (ii) tumor characteristics (cancer stage, histologic subtype, tumor grade, tumor size, and lymph node status), (iii) treatment types (hysterectomy and radiotherapy), and (iv) survival outcome (cause-specific survival (CSS) and overall survival (OS)).

### 2.3. Study Definition

The chronologic time sequence was examined between the uterine and colorectal cancer diagnoses, and when the time interval between the two diagnoses was ≤2 months, the cases were considered to be synchronous SPC as previously described [23]. SPC diagnosed >2 months after the index cancer was considered postcedent SPC, whereas SPC diagnosed >2 months prior to the index cancer was considered antecedent SPC.

Recorded cancer stages were based on the American Joint Committee on Cancer surgical-pathological staging classification schema [24]. The International Classification of Diseases for Oncology third edition site/histology validation and the World Health Organization histological classification codes were used to group histologic subtypes as described before [21,22,25]. The grouping and categorization of clinico-pathological factors for this study were based on prior studies [21,22,25]. CSS was defined as the time between uterine cancer diagnosis and death from uterine cancer. OS was defined as the time between uterine cancer diagnosis and death from any cause. Cases without survival events were censored at the last follow-up. Cause of death is linked with state mortality records and the National Death Index [26].

### 2.4. Study Aim

The primary objective of the study was to examine temporal trends of secondary colorectal cancer in women with uterine cancer. The secondary objective was to examine clinico-pathological characteristics and outcomes of women with uterine cancer who developed secondary colorectal cancer. The timing of colorectal cancer diagnosis relative to uterine cancer diagnosis (antecedent, synchronous, and postcedent) among those who had two cancer diagnoses was examined and compared to those without secondary colorectal cancer.

### 2.5. Statistical Consideration

Continuous variables were assessed for normality with the Kolmogorov–Smirnov test, and differences were assessed by the one-way ANOVA or Kruskal–Wallis *H* test as appropriate. Differences in ordinal or categorical variables were assessed by the chi-square test. Cox proportional hazard regression models were used to estimate hazard ratio (HR) with 95% confidence interval (CI) for multivariable analysis [27]. Patient demographics, tumor characteristics, and treatment types were entered in the final model.

Joinpoint Trend Software (version 4.4.0.0, National Cancer Institute, Bethesda, MD, USA) was used to identify the temporal changes in the trend analysis [28]. Time increments were grouped by every one calendar year to provide percent frequencies with CIs or every one year of age to provide mean values with standard errors of collected variables. The linear segmented regression test was used for the analysis, and log-transformation was performed to determine annual percent change of the slope with 95% CI [29].

For a sensitivity analysis, incidence of postcedent colorectal cancer was examined among uterine cancer women without antecedent or synchronous colorectal cancer who did not die of uterine cancer and had a follow-up of ≥5 (and ≥10) years, termed as cancer survivors in this study. Among the survivors, trends in standardized incidence ratio (SIR) for uterine cancer and colorectal cancer were also examined based on observed and expected events per standardized population [30]. This is based on the rationale that the diagnosis of postcedent SPC is dependent on follow-up time, and a short follow-up interval may result in lead-time bias for SPC [31]. Trends in SPC-specific mortality were examined among survivors. The study population with uterine cancer was also limited to only endometrial tumors, as the majority of uterine cancer is epithelial, particularly endometrioid-type. Moreover, temporal trends of secondary uterine cancer were examined in the colorectal cancer cohort.

All analyses were based on two-tailed hypotheses, and a *p* < 0.05 was considered statistically significant. Statistical Package for Social Sciences (SPSS, version 24.0, IBM Corp, Armonk, NY, USA) was used for the statistical analysis. The STROBE guidelines were outlined for performance of this observational study [32].

## 3. Results

The study schema is shown in Appendix A. There were 6862 women with both cancer diagnoses, representing 2.8% (95% CI 2.7–2.9) of the uterine cancer cohort (*n* = 246,272) and 1.6% (95% CI 1.6–1.7) of the colorectal cohort (*n* = 421,312). Among 6811 women with uterine cancer in whom time to secondary colorectal cancer was available, postcedent colorectal cancer after uterine cancer was most common (*n* = 4093 (60.1%), median time interval 7.6 years (interquartile range (IQR), 3.4–14.1)), followed by antecedent colorectal cancer prior to uterine cancer (*n* = 2103 (30.9%), median time interval 4.9 years (IQR, 2.2–9.5)) and synchronous uterine and colorectal cancers (*n* = 615, 9.0%).

In the uterine cancer cohort, women with antecedent colorectal cancer had similar characteristics to those with synchronous colorectal cancer and were more likely to be older than those without secondary colorectal cancer (*p* < 0.001; Table 1). Women with postcedent colorectal cancer were similar to those without secondary colorectal cancer and were more likely to be white, married and to have received radiotherapy (all, *p* < 0.001). Uterine tumors in the postcedent colorectal cancer group were more likely to be endometrioid type, early-stage, and low-grade but less likely to have nodal involvement, whereas tumors in the antecedent and synchronous colorectal cancer groups were more likely to have high-grade histology and be advanced-stage compared to tumors without secondary colorectal cancer (all, *p* < 0.01).

Age at uterine cancer diagnosis was similar between those with and without secondary colorectal cancer until the mid-1980s in the uterine cancer cohort (mean, 59.2 versus 60.8 in 1973; *p* = 0.15; Figure 1A); subsequently, women with secondary colorectal cancer were diagnosed at older ages compared to those without secondary colorectal cancer (66.9 versus 62.5 in 2013; *p* < 0.001). In the colorectal cancer cohort (Figure 1B), women who had secondary uterine cancer were younger compared to those without secondary uterine cancer until the early-1990s (62.8 versus 68.8 in 1973; *p* = 0.005); however, afterwards, women who had secondary uterine cancer were older compared to those without secondary uterine cancer (69.5 versus 67.9 in 2013; *p* = 0.057).

The number of women with uterine cancer who had secondary colorectal cancer decreased from 4.5% to 1.2% between 1992–2013 (relative risk reduction (RRR) 74.0%; *p* < 0.001; Figure 2A). When stratified by the chronology of colorectal cancer diagnoses, the number of women with uterine cancer with synchronous colorectal cancer decreased from 0.5% to 0.2% between 1979–2013 (RRR 56.2%, *p* < 0.001). Contrastingly, there was an increase in the number of women with uterine cancer with antecedent colorectal cancer from 0.6% to 0.9% between 2001–2013 (45% relative risk increase (RRI), *p* = 0.014). The number of women with colorectal cancer who had secondary uterine cancer increased from 0.8% to 2.0% between 1973–1986 (1.5-fold RRI, *p* < 0.001; Figure 2B). When stratified by the chronology of diagnoses, the number of colorectal cancer women who had antecedent uterine cancer increased from 0.8% to 1.3% between 2001–2013 (66% RRI, *p* < 0.001).

Trends of postcedent colorectal cancer were examined among 123,940 uterine cancer who survived ≥5 years (median follow-up 11.7 years, IQR 8.0–17.9; Appendix A). The number of uterine cancer survivors who developed postcedent colorectal cancer decreased from 5.3% to 0.7% between 1981–2008 (RRR 87.0% *p* < 0.001; Figure 3A). Similar results were observed among the 75,624 women with uterine cancer who survived ≥10 years (median follow-up 16.0 years, IQR 12.4–17.1), and RRR was 79.2% between 1982–2003 (*p* < 0.001). When standardized by the colorectal cancer incidence in the general population (Figure 3B), postcedent colorectal cancer decreased among ≥5-year uterine cancer survivors between 1975–2008 (RRR 80.5%, *p* < 0.005) and among ≥10-year uterine cancer survivors between 1975–2003 (RRR 72.0%, *p* < 0.001). The decreased incidence of postcedent colorectal cancer was particularly observed in the group diagnosed with colorectal cancer ≥5 years after uterine cancer diagnosis (Appendix A).

Among 141,801 women with colorectal cancer who survived ≥5 years (median follow-up 10.3 years, IQR 7.3–14.3; Appendix A), the number of colorectal cancer survivors who developed postcedent uterine cancer decreased from 1.7% to 0.5% between 1973–2008 (RRR 71.6%, *p* < 0.001; Figure 3A). Similarly, among 73,582 women who survived ≥10 years (median follow-up 14.1 years, IQR 11.8–19.1), the RRR of postcedent uterine cancer was 58.2% between 1973–2003 (*p* < 0.001). When standardized by the uterine cancer incidence in the general population (Figure 3B), postcedent uterine cancer decreased among ≥5-year colorectal cancer survivors between 1986–2008 (RRR 78.7%, *p* < 0.001) and among ≥10-year uterine cancer survivors between 1986–2003 (RRR 66.2%, *p* < 0.001). The decreased incidence of postcedent uterine cancer was also observed particularly in the group diagnosed with uterine cancer ≥5 years after colorectal cancer diagnosis (Appendix A).

The age-specific trends of secondary colorectal cancer among uterine cancer women were examined, stratified by the chronology of the colorectal cancer diagnoses (Appendix AA). After age 60, the number of women with uterine cancer with secondary colorectal cancer increased (*p* < 0.01). The peak age for antecedent colorectal cancer was 88 compared to 75 for postcedent colorectal cancer (both, *p* < 0.001). A trend of increasing synchronous colorectal cancer was observed after age 54 (*p* < 0.001). Age-specific trends of secondary uterine cancer among women with colorectal cancer were also examined (Appendix AB). The peak age of secondary uterine cancer diagnosis overall was 66 (*p* < 0.05), contrasting to the peak age of diagnosis for antecedent uterine cancer only, which was 79 years (*p* < 0.005). A trend of decreasing synchronous and postcedent uterine cancer diagnosis was observed after the mid-60s (66 and 65, respectively; *p* < 0.001).

The relative time interval between the two cancer diagnoses was examined among the 6811 women with both diagnoses. Uterine cancer diagnosis was more likely to antecede colorectal cancer diagnosis after age 40 (Figure 4A), whereas colorectal cancer was more likely to antecede uterine cancer after age 76 (Figure 4B) (both, *p* < 0.001). The median absolute time interval between the two diagnoses was 5.8 years (IQR, 2–11.5). The absolute time interval between the two diagnoses was shorter when uterine cancer was diagnosed at an older age, whereas the time interval was longer when colorectal cancer was diagnosed at older age (both, *p* < 0.001; Appendix A).

In the uterine cancer cohort, the median follow-up of censored cases was 6.6 years, and there were 110,219 all-cause deaths including 44,811 uterine cancer deaths during the follow-up. Irrespective of the chronology of secondary colorectal cancer diagnosis, uterine cancer women who had secondary colorectal cancer had superior CSS compared to those who did not have secondary colorectal cancer on multivariable analysis (adjusted-HR: 0.82 for antecedent, 0.77 for synchronous, and 0.26 for postcedent, all *p* < 0.01; Table 2).

In contrast, women who had antecedent or synchronous colorectal cancer had poorer OS (adjusted-HR: 1.26 for antecedent, and 1.55 for synchronous, both *p* < 0.001) whereas those who had postcedent colorectal cancer had superior OS (adjusted-HR: 0.91, *p* < 0.001) compared to the non-SPC group. The number of uterine cancer survivors who died of colorectal cancer decreased from 2.2% to <0.1% between 1982–2008, and the number of colorectal cancer survivors who died of uterine cancer decreased from 0.3% to <0.1% between 1997–2006 (both, *p* < 0.001; Appendix A).

Histology-specific analyses were performed in 227,275 women with endometrial cancer and in 177,897 women with endometrioid-type endometrial cancer. In both cohorts, patient characteristics were similar to the entire study cohort, and postcedent colorectal cancer also decreased among women with long-term follow-up during the study period (Appendix A and Appendix A). Like the whole cohort, CSS was also superior in women with secondary colorectal cancer compared to those without secondary colorectal cancer, whereas OS was inferior in women with antecedent or synchronous colorectal cancer in the two subgroups (Table 2).

## 4. Discussion

This study found that secondary colorectal cancer among uterine cancer is not rare, and ~3% of women with uterine cancer developed secondary colorectal cancer. Previously, studies have shown an increased risk of secondary colorectal cancer in women with uterine cancer; however these studies did not evaluate trends in timing or chronology of the two cancer diagnoses [4,5,6,7,8,9]. Our study is unique in showing that there is a recent decrease in the number of women with uterine cancer who had secondary colorectal cancer, which is attributable to a decrease in postcedent colorectal cancer following uterine cancer.

Various causalities may explain the recent decrease in postcedent colorectal cancer after uterine cancer diagnosis. First, successful implementation of colorectal cancer screening strategies in the United States may be identifying more pre-cancerous lesions and reducing the incidence of subsequent colorectal cancer [17,18], and this may also apply to women with uterine cancer. In 2015, nearly two thirds of U.S. adults aged 50–75 received colorectal cancer screening, and the population-based incidence of colorectal cancer has decreased from 57 to 37 per 100,000 between 1992–2015 (RRR 35.1%) [33,34,35]. Because diagnosis of uterine cancer is likely to be antecedent to colorectal cancer among those eventually diagnosed with both malignancies, as shown both in this study and others [4], age-based colorectal cancer screening may be warranted in those with uterine cancer even without Lynch syndrome.

Second, increasing utilization of genetic testing for women with both uterine and colorectal cancers may be responsible for decreasing rates of postcedent uterine cancer and colorectal cancer cases. Universal Lynch syndrome screening has been suggested for women with both colorectal and uterine cancers, given high prevalence in this unique population. A recent nationwide survey of genetic counselors showed significant increases in the utilization of routine Lynch syndrome screening for newly diagnosed colorectal and uterine cancer patients [36]. In women with Lynch syndrome, surveillance strategies such as frequent colonoscopies to identify pre-cancerous lesions or possible chemoprevention with aspirin may also have contributed to the decrease in postcedent colorectal cancer [37]. It would be of interest to examine whether the decreasing trends in postcedent uterine cancer after colorectal cancer in women with Lynch syndrome may be due to increasing utilization of genetic testing and recommendations for risk-reducing hysterectomy [14,38].

Third, changes in treatment patterns for uterine cancer may have led to the decrease in postcedent colorectal cancer after uterine cancer. There has been a decrease in the utilization of whole pelvic radiotherapy with a concomitant increase in vaginal brachytherapy in surgically treated women with uterine cancer [39]. As pelvic irradiation may increase the risk of colorectal cancer [11], the decrease in whole pelvic radiotherapy use may have indirectly resulted in decreasing postcedent colorectal cancer incidence after uterine cancer.

Another salient finding of this study is that there has been a recent increase in antecedent SPC in both uterine and colorectal cancers. Parallel to this trend, women in the SPC group became older, likely reflecting our aging society (~15% of the North American population was ≥65 years in 2015), resulting in more occasions to develop another malignancy during follow-up [15].

Our study demonstrated that patient demographics, tumor characteristic, and survival of women with uterine cancer are largely different based on the chronology of secondary colorectal cancer, potentially implying different disease characteristics or patient factors in antecedent and synchronous (more aggressive) versus postcedent (less aggressive) SPCs. Women with postcedent colorectal cancer were younger, which is likely responsible for the superior OS in this group compared to other types of SPC. Additionally, whether Lynch syndrome may be associated with postcedent colorectal cancer is of interest and deserves further exploration, especially given that Lynch syndrome-related uterine cancer is more likely to be diagnosed at a young age and have favorable histology [14].

Limitations of this study include that this is a retrospective study and there may be missing confounders for analysis. For instance, no information was available for patient’s geographic relocation between the data-covered and uncovered areas, thus some cases of SPC may not be captured by the database, resulting in misclassification. Information for genetic testing is also not available in the datasets, and we do not know whether mutations of genes predispositions co-occur in both cancers or are mutually exclusive and whether such mutations have any contribution towards chronology of SPC.

Another limitation of our study may be inadequate duration of follow-up to capture SPC, which may have contributed to the decreased numbers of postcedent SPC diagnoses in more recent years. A recent large-scale analysis of nearly three million malignancies found that tumor registries with shorter operating times (<10 years) are less likely to capture SPC compared to registries spanning greater lengths of time [31]. The current database includes over 40 years of operation, and we further examined patients who had long-term follow-up for analysis (Appendix A). Moreover, we observed similar decreasing trends of postcedent SPC in the two cohorts when examined with a one-year latency exclusion period in a post-hoc analysis (data not shown) [40]. Thus, both the use of this database, which has been previously validated and spans several decades, and the consistency of our results in long-term cancer survivors, support our findings of a recent decrease in postcedent colorectal cancer in women with uterine cancer as well as decreasing incidence of postcedent uterine cancer in women with colorectal cancer.

## 5. Conclusions

In summary, our data likely reflect demographic changes both in the U.S. population as well as in women with uterine cancer and colorectal cancer. In particular, the incidence of secondary colorectal cancer has decreased among women with uterine cancer, especially colorectal cancer diagnosed after uterine cancer. Similar trends of decreasing postcedent uterine cancer among colorectal cancer women were also observed. These results are encouraging for care providers and patients, and further study to validate these findings in a different population is warranted.

## Figures and Tables

**Figure 1 jcm-08-00714-f001:**
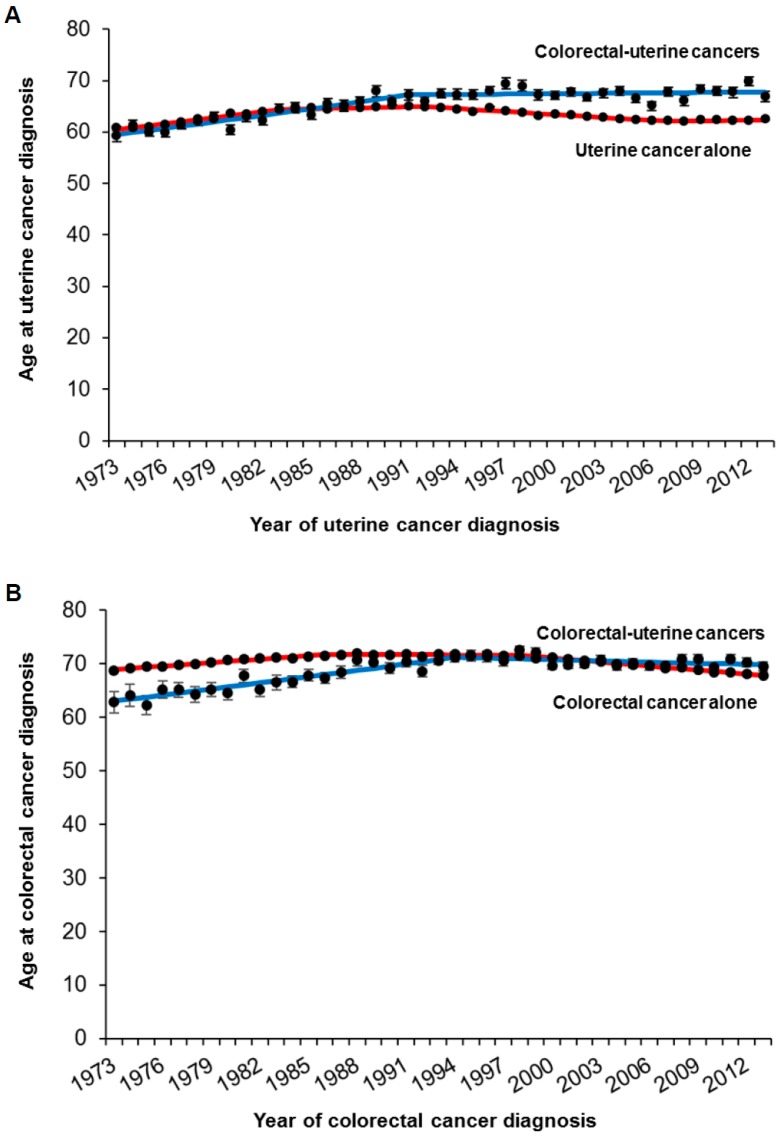
Trends in age at diagnosis. Age at diagnosis of the index cancer is shown for (**A**) uterine cancer cohort and (**B**) colorectal cancer cohort. Dots represent actual mean values. Bars represent standard errors. Colored lines represent modeled lines. Raw results of temporal trends are shown in Appendix A.

**Figure 2 jcm-08-00714-f002:**
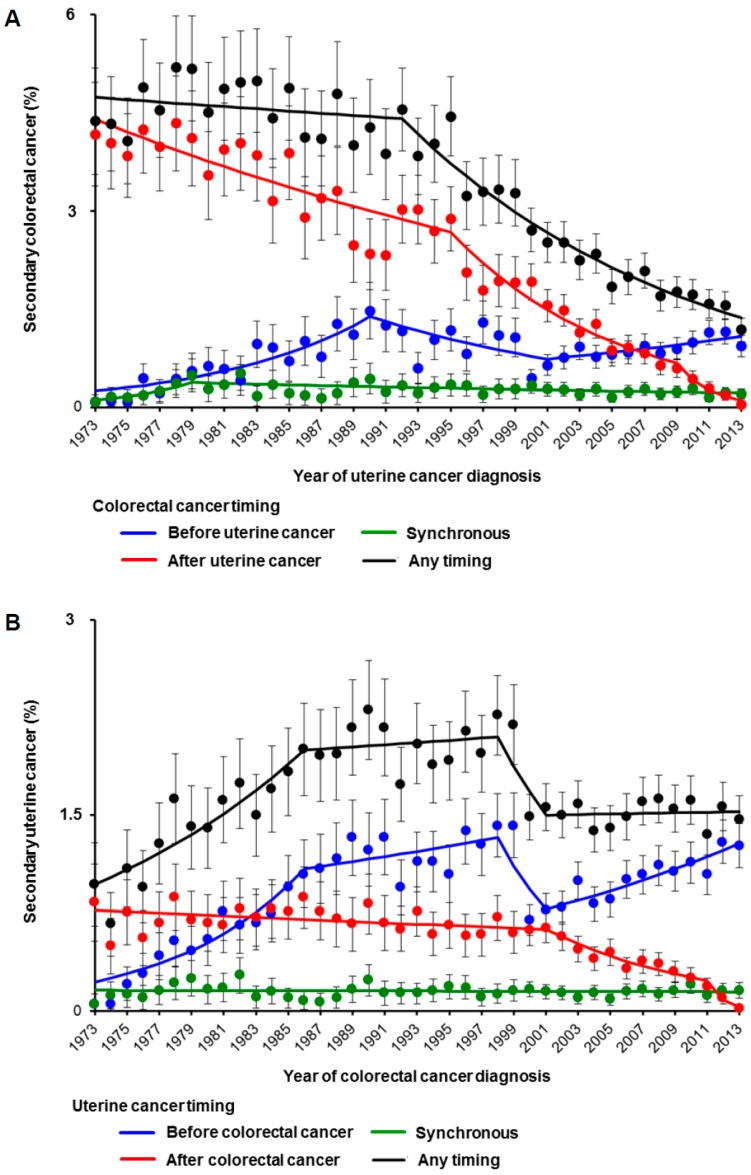
Trends in secondary primary cancer per year. Temporal trends are shown for (**A**) secondary colorectal cancer among uterine cancer and (**B**) secondary uterine cancer among colorectal cancer based on the timing of secondary primary cancer. Dots represent actual observed values. Bars represent standard confidence intervals. Colored lines represent modeled lines. Raw results of temporal trends are shown in Appendix A.

**Figure 3 jcm-08-00714-f003:**
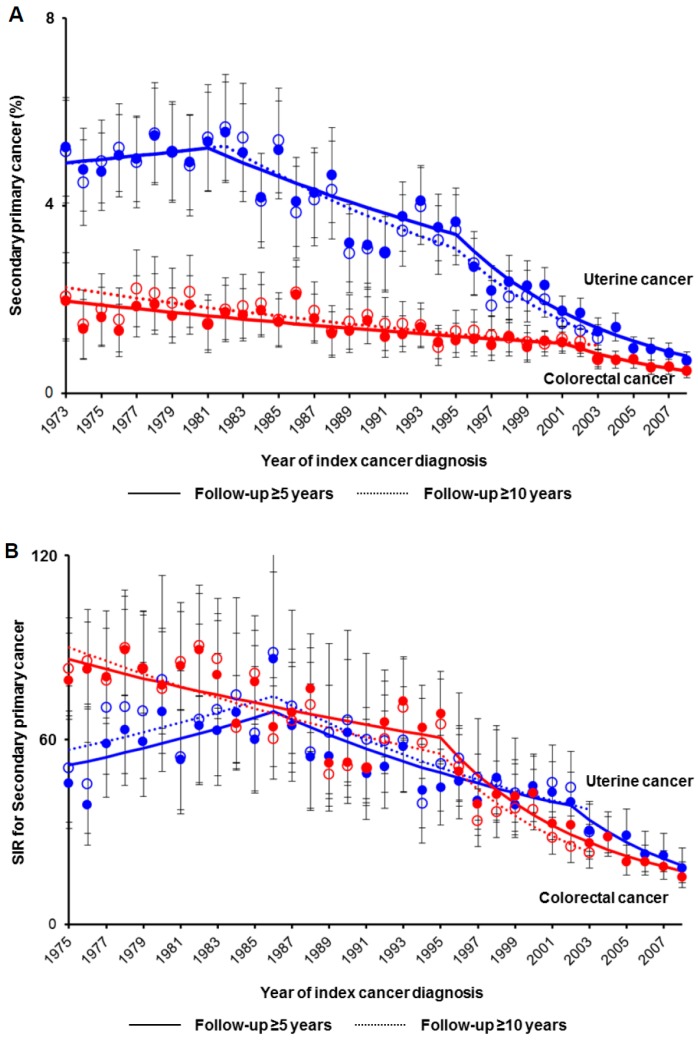
Trends of postcedent secondary primary cancer among cancer survivors per year. Among women who were alive at a follow-up ≥5 years (solid lines) or ≥10 years (dashed lines), (**A**) temporal trends of the incidence of postcedent secondary cancer and (**B**) standardized incidence ratio were examined. Dots represent actual observed values. Bars represent standard confidence intervals. Colored lines represent modeled lines. Blue lines represent the uterine cancer cohort, and red lines represent the colorectal cancer cohort, respectively. Raw results of temporal trends are shown in Appendix A. Median follow-up times across the study period are shown in Appendix A. Abbreviation: SPC, secondary primary cancer; and SIR, standardized incidence ratio.

**Figure 4 jcm-08-00714-f004:**
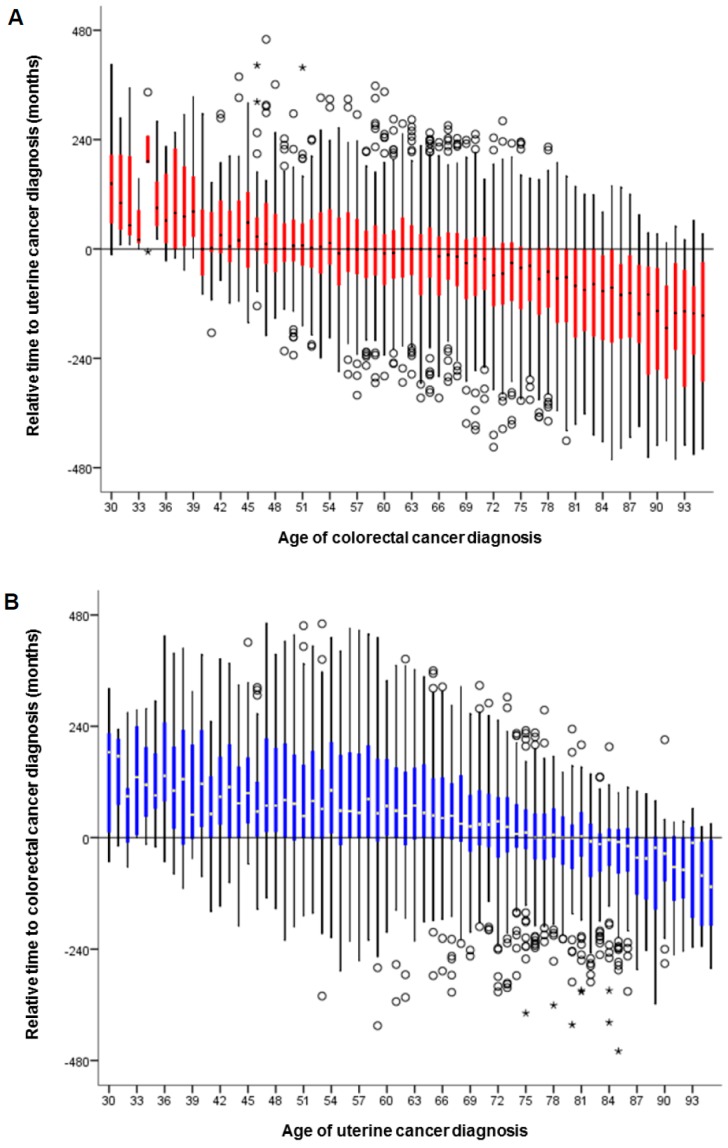
Relative time to secondary primary cancer per age. Among 6811 women who had the two cancer diagnoses, relative time to secondary primary cancer is displayed per age at the index cancer diagnosis. (**A**) Relative time to uterine cancer diagnosis is shown per age of colorectal cancer diagnosis. The first age that the median relative time interval became zero was 40 years. (**B**) Relative time to colorectal cancer diagnosis is shown per age of uterine cancer diagnosis. The first age that the median relative time interval became zero was 76 years.

**Table 1 jcm-08-00714-t001:** Patient demographics for uterine cancer based on timing of secondary colorectal cancer.

Characteristic	No CRC	Antecedent-CRC	Synchronous-CRC	Postcedent-CRC	*p*-Value
Number	*n* = 239,410	*n* = 2103	*n* = 615	*n* = 4093	
Age	62.9 (± 12.6)	70.5 (± 12.4)	68.5 (± 12.4)	63.1 (± 11.5)	**<0.001**
Year					**<0.001**
Before 1980	20,564 (8.6%)	59 (2.8%)	51 (8.3%)	883 (21.6%)	
1980–1989	27,218 (11.4%)	238 (11.3%)	80 (13.0%)	978 (23.9%)	
1990–1999	39,864 (16.7%)	448 (21.3%)	121 (19.7%)	987 (24.1%)	
2000–2009	100,853 (42.1%)	814 (38.7%)	252 (41.0%)	1125 (27.5%)	
2010 or later	50,911 (21.3%)	544 (25.9%)	111 (18.0%)	120 (2.9%)	
Race/ethnicity					**<0.001**
White	184,497 (77.1%)	1661 (79.0%)	499 (81.1%)	3445 (84.2%)	
Black	18,661 (7.8%)	153 (7.3%)	58 (9.4%)	238 (5.8%)	
Hispanic	18,993 (7.9%)	137 (6.5%)	32 (5.2%)	170 (4.2%)	
Asian	12,611 (5.3%)	121 (5.8%)	21 (3.4%)	189 (4.6%)	
Others *	4648 (1.9%)	31 (1.5%)	5 (0.8%)	51 (1.2%)	
Area					**<0.001**
West	123,311 (51.5%)	1021 (48.5%)	260 (42.3%)	1966 (48.0%)	
Central	55,898 (23.3%)	584 (27.8%)	168 (27.3%)	1195 (29.2%)	
East	60,201 (25.1%)	498 (23.7%)	187 (30.4%)	932 (22.8%)	
Marital status					**<0.001**
Single	35,108 (14.7%)	205 (9.7%)	87 (14.1%)	475 (11.6%)	
Married	123,912 (51.8%)	945 (44.9%)	293 (47.6%)	2290 (55.9%)	
Others *	80,390 (33.6%)	953 (45.3%)	235 (38.2%)	1328 (32.4%)	
Histology types					**<0.001**
Endometrioid	172,784 (72.2%)	1416 (67.3%)	416 (67.6%)	3250 (79.4%)	
Serous	13,975 (5.8%)	173 (8.2%)	55 (8.9%)	221 (5.4%)	
Clear cell	2938 (1.2%)	45 (2.1%)	13 (2.1%)	41 (1.0%)	
Carcinosarcoma	10,664 (4.5%)	159 (7.6%)	28 (4.6%)	84 (2.1%)	
Sarcoma	10,376 (4.3%)	44 (2.1%)	20 (3.3%)	69 (1.7%)	
Mixed	7318 (3.1%)	95 (4.5%)	17 (2.8%)	45 (1.1%)	
Others	21,357 (8.9%)	171 (8.1%)	66 (10.7%)	383 (9.4%)	
Stage					**<0.001**
I	150,210 (62.7%)	1263 (60.1%)	326 (53.0%)	2525 (61.7%)	
II	10,034 (4.2%)	119 (5.7%)	33 (5.4%)	172 (4.2%)	
III	20,446 (8.5%)	221 (10.5%)	64 (10.4%)	219 (5.4%)	
IV	18,163 (7.6%)	165 (7.8%)	65 (10.6%)	111 (2.7%)	
Unknown	40,557 (16.9%)	335 (15.9%)	127 (20.7%)	1066 (26.0%)	
Grade					**<0.001**
1	83,271 (34.8%)	538 (25.6%)	174 (28.3%)	1514 (37.0%)	
2	62,105 (25.9%)	559 (26.6%)	163 (26.5%)	1307 (31.9%)	
3	51,833 (21.7%)	609 (29.0%)	149 (24.2%)	666 (16.3%)	
Unknown	42,201 (17.6%)	397 (18.9%)	129 (21.0%)	606 (14.8%)	
Tumor size					**<0.001**
≤2 cm	21,820 (9.1%)	205 (9.7%)	47 (7.6%)	284 (6.9%)	
>2 cm	81,395 (34.0%)	745 (35.4%)	185 (30.1%)	893 (21.8%)	
Unknown	136,194 (56.9%)	1153 (54.8%)	383 (62.3%)	2916 (71.2%)	
Pelvic lymph node ^†^					**<0.001**
Not involved	167,161 (91.6%)	1444 (91.7%)	402 (92.0%)	2439 (95.6%)	
Involved	15,236 (8.4%)	131 (8.3%)	35 (8.0%)	112 (4.4%)	
Lymph node ratio (%) ^†^	20.0 (9.1-50.0)	33.3 (12.5-57.1)	31.3 (17.0-100)	18.6 (10.0-38.2)	**0.004**
Hysterectomy					**<0.001**
No	19,763 (8.3%)	306 (14.6%)	64 (10.4%)	102 (2.5%)	
Yes	187,201 (78.2%)	1657 (78.8%)	451 (73.3%)	2744 (67.0%)	
Unknown	32,446 (13.6%)	140 (6.7%)	100 (16.3%)	1247 (30.5%)	
Radiotherapy					**<0.001**
None	165,520 (69.1%)	1534 (72.9%)	470 (76.4%)	2486 (60.7%)	
External beam	48,940 (20.4%)	358 (17.0%)	108 (17.6%)	1117 (27.3%)	
Implants	13,082 (5.5%)	130 (6.2%)	19 (3.1%)	138 (3.4%)	
Both	6861 (2.9%)	32 (1.5%)	6 (1.0%)	290 (7.1%)	
Unknown	5007 (2.1%)	49 (2.3%)	12 (2.0%)	62 (1.5%)	

Number (percent per column), median (interquartile range), or mean (standard deviation) is shown. Significant *p*-values are emboldened. Cases without known time interval to secondary colorectal cancer were not included. * including unknown cases. ^†^ among staged cases with available results. Abbreviation: CRC, secondary colorectal cancer.

**Table 2 jcm-08-00714-t002:** Associations of secondary colorectal cancer and uterine cancer survival based on histology types.

		Cause-Specific Survival		Overall Survival	
Characteristic		HR (95% CI)	*p*-Value	HR (95% CI)	*p*-Value
All histology	Secondary colorectal cancer				
	No	1		1	
	Yes (any timing)	0.69 (0.67–0.72)	**<0.001**	1.01 (0.99–1.03)	0.08
	Secondary colorectal cancer				
	No	1		1	
	Antecedent	0.82 (0.74–0.91)	**<0.001**	1.26 (1.19–1.33)	**<0.001**
	Synchronous	0.77 (0.63–0.93)	**0.008**	1.55 (1.41–1.71)	**<0.001**
	Postcedent	0.26 (0.23–0.30)	**<0.001**	0.91 (0.88–0.94)	**<0.001**
Endometrial	Secondary colorectal cancer				
	No	1		1	
	Yes (any timing)	0.70 (0.67–0.73)	**<0.001**	1.02 (1.01–1.04)	**0.005**
	Secondary colorectal cancer				
	No	1		1	
	Antecedent	0.82 (0.74–0.92)	**0.001**	1.26 (1.19–1.33)	**<0.001**
	Synchronous	0.81 (0.66–0.99)	**0.046**	1.67 (1.5–1.85)	**<0.001**
	Postcedent	0.28 (0.24–0.32)	**<0.001**	0.93 (0.90–0.97)	**<0.001**
Endometrioid	Secondary colorectal cancer				
	No	1		1	
	Yes (any timing)	0.68 (0.63–0.70)	**<0.001**	1.04 (1.02–1.06)	**<0.001**
	Secondary colorectal cancer				
	No	1		1	
	Antecedent	0.78 (0.67–0.91)	**0.002**	1.32 (1.23–1.42)	**<0.001**
	Synchronous	0.65 (0.50–0.86)	**0.002**	1.55 (1.37–1.75)	**<0.001**
	Postcedent	0.29 (0.25–0.34)	**<0.001**	0.99 (0.95–1.03)	0.55

Multivariable analysis with Cox proportional hazard regression models for the results. Covariates adjusted for the association of secondary colorectal cancer and uterine cancer survival included patient age (continuous), Year at diagnosis (<1980, 1980–1989, 1990–1999, 2000–2009, and ≥2010), registry area (West, Central, and East), race/ethnicity (White, Black, Hispanic, Asian, and others), marital status (single, married, and others), histology (endometrioid, serous, clear cell, carcinosarcoma, sarcoma, others), cancer stage (I, II, III, IV, and unknown), tumor grade (1, 2, 3, and unknown), and tumor size (≤2, >2 cm, and unknown), hysterectomy (no, yes, and unknown), and radiotherapy (no, yes, and unknown). Abbreviations: HR, hazard ratio; and CI, confidence interval.

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
