# Peer review of "Decreasing Trends of Secondary Primary Colorectal Cancer among Women with Uterine Cancer: A Population-Based Analysis"

_jcm, 2019, doi:10.3390/jcm8050714_

Round 1
Reviewer 1 Report
In this research paper, the authors examined the trend, characteristics and outcomes of women with uterine cancer who had secondary colorectal cancer (CRC) using. Their finding suggest that secondary CRC among women who had uterine cancer is not rare and ~3% of women with uterine cancer developed secondary CRC. They discussed various possible reasons for this trend-in postcedent colorectal cancer after uterine cancer diagnosis. For example, implementation of a colorectal cancer screening strategies by identifying more pre-cancerous lesions, or increasing trend in genetic testing for both uterine and CRC or routine Lynch syndrome screening for newly diagnosed colorectal and uterine cancer patients. Alternative possibility for the decreased trend include a decrease in the utilization of whole pelvic radiotherapy and an increase in vaginal brachytherapy in surgically-treated women with uterine cancer. Their analysis suggests that incidence of secondary CRC has decreased among women with uterine cancer.
Author Response
Reply:
We appreciate the reviewer’s comment and remark.
Reviewer 2 Report
The paper is well written and represents a valuable work from an epidemiological point of view; Data curation seems precise and analyzes performed extensive, including sensitivity analysis to confirm robustness of the findings.
Author Response

(The authors gave the same response as above.)

Reviewer 3 Report
Authors should elaborate on the introduction part a bit more emphasizing genetic predisposition and family history of uterine and colorectal cancer. Whether mutations of genes responsible for HNPCC co-occur in both types of cancers or are mutually exclusive and has any contribution towards antecedent or postcedent of uterine and/or colorectal cancer.
Author Response
Reply:
We respectfully acknowledged the reviewer's comment about genetic predisposition. While we agree with reviewer to discuss about this in the Introduction section, given lack of information for genetic testing result in our study, we feel that it will fit more in limitation section in the Discussion. Per the suggestion, we have made modifications accordingly in the revised manuscript.